# Drug use in street sex workers (DUSSK) study protocol: a feasibility and acceptability study of a complex intervention to reduce illicit drug use in drug-dependent female street sex workers

Nicola Jeal,[1,2] Rita Patel,[2,3] Niamh M Redmond,[2,3] Joanna M Kesten,[2,3,4] Sophie Ramsden,[1] John Macleod,[2,5] Joanna Coast,[3,6] Maggie Telfer,[7] David Wilcox,[8] Gill Nowland,[9] Jeremy Horwood[3,5]

For numbered affiliations see end of article.

**Correspondence to**
Dr Nicola Jeal;
nikki.jeal@bristol.ac.uk

## ABSTRACT

**Introduction** Poor health of sex workers continues to be a source of international concern. Sex work is frequently linked with problematic drug use and drug-dependent sex workers typically work on the street, experiencing the greatest risks to health compared with the general population. Street sex workers (SSWs) are much more likely to have experienced incidences of physical and sexual assault, increasing their risk of developing post-traumatic stress disorder (PTSD). We have developed a novel complex intervention designed to reduce illicit drug use in drug-dependent female SSWs which involves: female SSW drug treatment groups (provided by a specialist charity) in a female SSW setting (female sex worker charity premises) provided by female-only staff, PTSD care with eye movement desensitisation and reprocessing (EMDR) therapy provided by female staff from National Health Service (NHS) mental health services.

**Methods and analysis** A mixed methods study investigating the feasibility and acceptability of this intervention to inform the design of a future randomised controlled trial. The study aims to recruit up to 30 participants from November 2017 to March 2018 at a single site, with the intervention being delivered until December 2018. It will gather quantitative data using questionnaires and group attendance. Drug treatment group observations and in-depth interviews undertaken with up to 20 service users and 15 service providers to examine experiences and acceptability of the intervention. Study feasibility will be assessed by evaluating the recruitment and retention of participants to the intervention; the feasibility of NHS and third sector organisations working closely to coordinate care for a SSW population; the potential for specialist NHS mental health services to screen and provide EMDR therapy for drug-dependent SSWs and potential costs of implementing the intervention.

**Ethics and dissemination** This study was approved by South West–Frenchay Research Ethics Committee (REC reference: 17/SW/0033; IRAS ID: 220631) and the Health

## Strengths and limitations of this study

► This is a mixed methods study to investigate the feasibility and acceptability of a novel intervention designed to reduce levels of post-traumatic stress disorder in order to support a reduction in illicit drug use in female drug-dependent street sex workers (SSWs).

► The complex intervention addresses issues highlighted by female SSWs in previous qualitative work as well as quantitative systematic review evidence.

► The involvement of service users and a range of multidisciplinary service providers has been crucial in the development and design of the proposed intervention and study.

► Conclusions about effectiveness or efficacy of the intervention are not possible due to the study being a single-arm feasibility study; however, this study will enable the refinement of the intervention for a future effectiveness trial.

Research Authority (HRA). Findings will be disseminated through research conferences and peer-reviewed journals.

## INTRODUCTION

Sex workers are internationally recognised as a group who experience poor health.[1 2] Sex work and drug use are frequently linked,[3–5] and previous research has shown that street sex workers (SSWs) experience worse health than sex workers in off-street settings[6] and use heroin and crack cocaine as their main drugs of dependency.[7] Dependency on illicit drugs underpins their excess morbidity,[7 8] drives risk-taking while selling sex,[9 10] as well as the direct and indirect health risks of injection drug use.[11 12] Furthermore, illicit drug dependency can keep women entrenched in sex work as ceasing sex

work is inversely related to levels of injection drug use[13] and drug-dependent SSWs describe being trapped in a work-score-use cycle.[14]

Despite these significant drug treatment needs, drug-dependent SSWs have poorer outcomes from drug treatment services compared with other service users.[15 16] Previous SSW-focused interventions aiming to reduce levels of drug use have focused on heroin and/or crack cocaine and employed educational approaches,[9 17] substitute prescribing-based[18 19] and psychological approaches including motivational interviewing[20] but none convincingly demonstrated a positive effect in reducing drug use.[21] While the challenges of mixed-gender drug treatment services contribute to the lack of effectiveness[22 23] and cost-effectiveness for female service users in particular,[24] female SSWs have been found to face additional obstacles in mixed-gender groups related to their sex work history.[25] For example, feelings of stigmatisation from other male and female service users following disclosure of sex work and adverse interactions with previously known male service users potentially prevent SSWs from discussing unresolved trauma, undermining their engagement in treatment.

High levels of poor mental health, a significant problem among SSWs,[26 27] have previously been highlighted as contributing to poor drug treatment outcomes.[28] Experience of abuse and violence, common among SSWs,[29 30] has led to recommendations for female-only trauma-focused drug treatment interventions[31] and there is some evidence that certain subgroups, such as SSWs,[32] may benefit from a trauma-focused approach. A recent Cochrane review of treatment of comorbid post-traumatic stress disorder (PTSD) and drug dependency[33] suggested that individual trauma-focused therapy alongside drug treatment appeared to have best outcomes for PTSD and reducing levels of drug use in the longer term. However, groups with severe and complex presentations were excluded from most included studies and, to date, there is no robust evidence of the impact of an integrated treatment approach in female drug-dependent SSWs.

## DEVELOPMENT OF THE INTERVENTION
A novel intervention addressing the unique and complex needs of female drug-dependent SSWs was developed in collaboration with service providers and informed by existing research. It was designed to occur prior to typical 'mainstream' drug treatment interventions (for both male and female drug-dependent individuals) and proposes an integrated care pathway through an innovative multiagency partnership. This pathway includes:
1. Female SSW-only groups in an SSW-only environment facilitated by female members of local drug treatment services.
2. Screening for PTSD by female staff from local specialist National Health Service (NHS) mental health services.

3. One-to-one PTSD therapy (eye movement desensitisation and reprocessing (EMDR)) with a female NHS clinician working within a specialist trauma service.

Addressing sex working history during initial drug treatment groups as well as screening and treating underlying PTSD is designed to prepare SSWs to engage more effectively with mainstream drug services with the aim of achieving better long-term health outcomes. EMDR was selected as it is a recommended first-line treatment for PTSD in UK National Institute of Health and Care Excellence (NICE) guidelines,[34] and unlike cognitive behavioural therapy, it does not require homework which may be a challenge for drug-dependent SSW and can be a relatively short course of treatment (NICE guidelines recommend up to 12 sessions).[34] EMDR is a form of psychotherapy which uses eye movements or other forms of bilateral stimulation and has similarities with slow-wave sleep and its role in memory consolidation[35] to purportedly assist clients in processing distressing memories and beliefs.[36] The use of EMDR in this population is a novel approach and understanding of its use in terms of acceptability in drug-dependent participants, including opioid substitution treatment (OST) is limited.

### Aims
This feasibility study will address the unanswered intervention questions required for a future large-scale randomised controlled trial to determine the effectiveness and cost-effectiveness of a complex intervention to reduce levels of PTSD in order to support a reduction in illicit drug use in female drug-dependent SSWs. The specific feasibility study objectives are to:
► Evaluate the recruitment and retention of participants to the intervention.
► Investigate the feasibility of three services of differing statutory and non-statutory, clinical and non-clinical backgrounds working closely to provide a complex intervention for drug-dependent female SSWs.
► Examine the experience and acceptability of the intervention for SSWs and service providers.
► Explore costs associated with the intervention.

## METHODS AND ANALYSIS
### Study design
The study uses a single-site mixed methods approach to investigate the feasibility and acceptability of a novel complex intervention designed to reduce levels of PTSD in order to support a reduction in illicit drug use in female-drug dependent SSWs. The protocol was written in accordance with the Standard Protocol Items: Recommendations for Interventional Trials guidelines.[37] The study aims to recruit up to 30 participants from November 2017 to March 2018, with the intervention being delivered until December 2018.

### Study setting
The study will take place in an inner city setting in a large UK city. Recruitment, drug group sessions and PTSD

assessment will take place in a female-only sex worker charity's drop-in support service, where advice, health and general day-to-day support are provided.

### Inclusion and exclusion criteria

Participants are eligible for the study if they are female aged 18 years or older, sold sex on the street in the UK at least once a week in the last calendar month (or 3 out of the 4 previous weeks) and have used heroin and or crack cocaine at least once a week in the last calendar month (or 3 out of the 4 previous weeks).

Participants are excluded from the study if they do not identify as female gender, are under 18 and have not sold sex on the street in the UK and not used heroin or crack cocaine at least once a week in the last calendar month.

### Participant recruitment

Study promotional flyers will be left in organisations and services that SSWs are known to use, such as a SSW charity outreach van and drop-in support service, housing organisations, specialist drug and alcohol services. SSWs can make direct contact with the researcher via telephone (with an answerphone facility) or ask support staff to phone on their behalf. Researchers will also attend the SSWs drop-in support service to directly approach potential participants with a promotional flyer, as proposed by the SSW patient and public involvement (PPI) consultation group who recommended this as the best arrangement for them. Participants will also be recruited by word-of-mouth through SSWs who are aware of the study and have contacts who may want to take part (ie, via snowball sampling).

The researcher will conduct eligibility screening according to the inclusion/exclusion criteria either face-to-face or over the telephone. Participants meeting the inclusion criteria will be invited to provide fully informed, written consent to participate in the study at the time of screening if that is face-to-face, or at a meeting arranged after telephone screening. Baseline assessment will be completed for all consenting participants and includes self-report measures of illicit drug use, sex work frequency and PTSD symptoms experienced (see Data collection methods section). A preferred communication and study contact strategy will be agreed with each individual at the outset of their participation. For participants not meeting the inclusion criteria, screening data will remain anonymised for eligibility reporting purposes only.

### The intervention pathway

It is recognised that participant progression through the intervention is unlikely to be linear and that group allocation and reallocation will be sensitive to the needs of individuals and other group members. All service provider partners will participate in monitoring how individuals and the wider group(s) are responding to the various aspects of the intervention; for example, we may find that women respond well to female, sex worker-only drug groups and develop stability behaviour more quickly than

expected, in which case, we may move them through the intervention quicker. We expect the intervention to take approximately 23 weeks or 6 months. Individual participants will be supported on a case-by-case basis which will be dependent on their drug use, treatment and engagement with services. Figure 1 details the most linear route possible through the intervention.

### 'Getting started' drug treatment group

The group will take place in a female sex worker charity premises with a maximum of eight places. The aim of the 'Getting started' group is to enable participants to achieve a level of stability, to reduce fear and anxiety about engaging in a group setting, to get used to the format and level of disclosure expected, to explore what skills are needed to engage in a group, to experience the feelings people are left with after a group and to learn how to manage these. During the group facilitation, topics that will be routinely and regularly covered are maintaining boundaries, why a group setting is used, personal resilience/strengths and setting SMART (specific, measurable, achievable, relevant and time bound) goals. After attending four group meetings, if participants are perceived by the group facilitators as exhibiting evidence of life/drug use stability such as engagement and functioning in the group, positive interaction with group facilitators, regular OST, they will be offered transfer to the 'Preparation for Recovery' group. The group will be an open group and participants may attend irregularly, but regular attendance will be encouraged via weekly phone text message reminders. Those participants who are injecting opiates (heroin) and are not currently receiving OST will be encouraged to access an OST prescription and be signposted to local services.

### 'Preparation for recovery' drug treatment group

The group will take place in a female sex worker charity premises, and there will be a maximum of eight places. The aim of the 'Preparation for recovery' group is to focus participants on building relationships and connections in the group, looking at peoples' barriers to motivation for change, weighing up pros and cons of drug use, exploring triggers for using, helping people to manage difficult feelings and looking at support networks. This part of the programme mimics mainstream drug services and aims to prepare women for joining mixed-gender groups. The group consists of a rolling programme of eight sessions to support participants to continue managing drug use and use of an OST prescription.

Participants who attend three consecutive 'Preparation for Recovery' groups and are assessed by group facilitators to be achieving drug stabilisation will be offered screening for PTSD. Participants found to be currently experiencing PTSD symptoms will be offered inclusion in the 'Stabilisation' group in preparation for receiving treatment for their PTSD symptoms while continuing to attend the 'Preparation for Recovery' group. If a participant is found not to be experiencing PTSD symptoms,

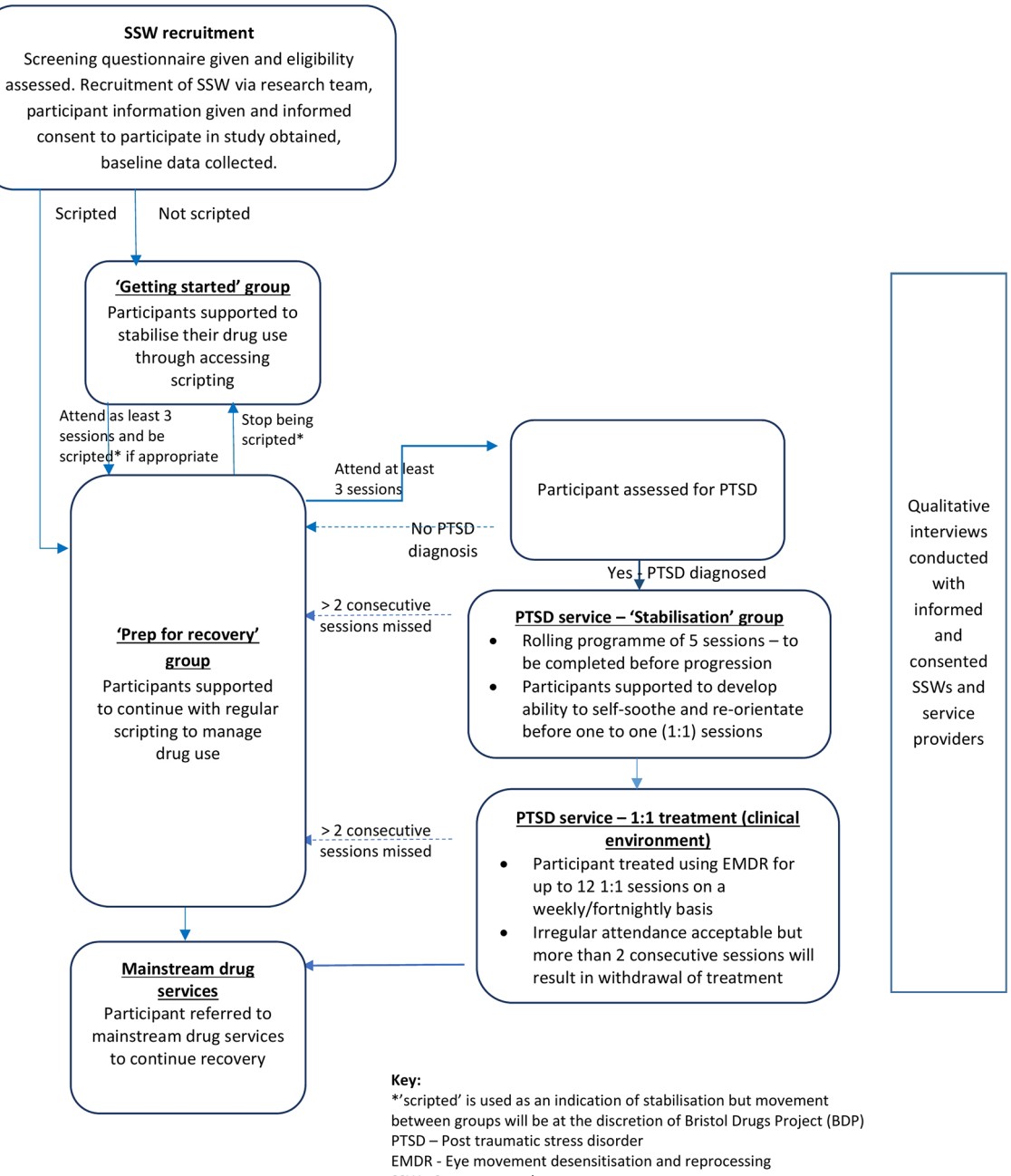

**Figure 1 DUSSK study** participant flow diagram.

they will continue in the 'Preparation for Recovery' group for the usual duration of the group (6–8 weeks) or until a group facilitator feels they are ready to be referred on to mainstream drug services. The group will be an open group and participants may attend irregularly, but regular attendance will be encouraged via phone text message reminders. If a participant is considered to no longer be achieving stability, they will be reassigned to the 'Getting started' group until they begin to stabilise once more.

### Screening for PTSD

Participants will be individually screened for currently experiencing PTSD symptoms in a 90 min one-to-one session with a registered female clinical psychologist. The

session will consist of a clinical interview to elicit information about symptoms related to the diagnostic criteria for PTSD as stated in the American Psychiatric Association's Diagnostic and Statistical Manual of Mental Disorders 5 (DSM-5).[38] The PTSD Check List-5 (PCL-5) is used to assist the clinical assessment, provide a baseline score and provide the clinical psychologist with a provisional PTSD diagnosis. If the participant is found to be currently experiencing PTSD symptoms, she will be offered a place in the 'Stabilisation' group. If she is deemed to not benefit from the 'Stabilisation' group, she can continue in the 'Preparation for recovery' group with eventual referral to mainstream drug services (see above).

## PTSD 'Stabilisation' group

The 'Stabilisation' group will be a rolling programme of five, 2-hour sessions to be held in a clinical setting. The aim of the group is to equip participants with the necessary skills to self-soothe and reorientate in preparation for the one-to-one EMDR treatment. The group will be facilitated by a female clinical psychologist. It is anticipated that the optimum size for the 'Stabilisation' group will be 3 to 12, and once all sessions have been completed, participants will be eligible to progress to one-to-one PTSD treatment. If participants fail to attend two consecutive sessions, they will be considered to have withdrawn from treatment.

## Treatment for PTSD

Treatment will occur in a clinical setting, for example, existing mental health service locations or local general practitioner (GP) practice private room. Participants will receive up to 12 one-to-one EMDR sessions for 90 min with a female clinical psychologist on a weekly, or fortnightly, basis. This treatment will aim to target the most distressing memories and process the dysfunctional information in order to reduce distress related to that memory and diminish the symptoms of PTSD experienced by participants. Aiming to improve self-esteem and self-efficacy, the treatment should enable the participant to better tolerate the residue of difficult experiences, reducing the need to self-medicate their distress with drugs and alcohol. If a participant is not sufficiently stabilised or too intoxicated due to recent drug or alcohol use, therapy will be deferred to a mutually agreed time and date. If participants continue to attend too intoxicated for treatment on two occasions, they will be discharged from EMDR treatment but can continue to attend 'Preparation for recovery' group sessions. If during treatment participants are assessed as experiencing acute symptoms that require further support from mental health services, the on-call crisis team will be contacted to arrange care. Participants' case workers will also be informed of the referral as part of ongoing support arrangements. Participants may attend EMDR sessions irregularly, but if they do not attend two consecutive appointments, they will be considered to have withdrawn from treatment. Regular attendance will be encouraged by reminders to attend appointments via text message, letter and case worker.

The participant will be encouraged to continue to be supported by the 'Preparation for Recovery' group during trauma screening, stabilisation group and one-to-one sessions. Women are able to see a clinical psychologist or request further assistance from mental health services in line with routine practice if required. Once the participant is assessed as having completed PTSD treatment, they will be referred by the 'Preparation for Recovery' group facilitators to attend mainstream drug services, to access ongoing support and treatment appropriate to their stage of recovery.

## Sample size

As this is a feasibility study, there is no formal sample size calculation. We aim to recruit up to 30 participants to fully evaluate the intervention processes and this is considered a large enough sample to estimate the proportion of eligible people who are willing to participate, attrition rate and consider the practicalities of recruitment and delivering the intervention.

## Data collection methods

### Baseline data collection

Baseline data will be collected by researchers at the time of recruitment once participant consent procedures are complete. Baseline data collected will be related to self-report of levels of illicit drug use, involvement in street sex work, completion of PCL-5[39] (a 20-item self-report measure that assesses the 20 DSM-5 symptoms of PTSD) and demographics (age, ethnicity).

### Process evaluation

Once recruited, attendance at and movement through the intervention will be monitored and recorded. Number of participants fully and partially completing the intervention pathway and patterns of attendance will be monitored weekly. An attendance register will be taken at the start of each of the groups by the group facilitators. Cost assessments in the feasibility study will be exploratory and informed by the qualitative interviews and service provider estimates for staff time and accommodation costs. These measures will also enable the assessment of the fidelity of the intervention as a whole against the proposed intervention pathway.

### Qualitative study

To examine participant and service provider views and experiences of the intervention, we will conduct observations (of the 'Getting started' and 'Preparation for Recovery' groups) to understand delivery, provide context and observe interactions and dynamics and undertake in-depth semistructured qualitative interviews to explore how the intervention could be made more acceptable and feasible. Qualitative findings will help to illuminate the strengths and weaknesses of the intervention and refine its final format.

With participants' verbal consent, a qualitative researcher will undertake up to 8 hours of non-participant observations to understand how the 'Getting started' and 'Preparation for Recovery' groups are operationalised and delivered in day-to-day practice. Any group member can ask the researcher to leave the group for any reason. If at any time the researcher's presence is considered by facilitators to be disrupting the group dynamics, they will leave the room. The researcher will write accounts of observations based on brief notes taken directly after the groups.[40] These field notes may include both direct observations and reflection on what has been observed. Observations will record activities, interactions and communication patterns.

All study participants will be asked at the time of study consent if they are willing to subsequently be contacted about taking part in a qualitative interview. A purposive sample of those agreeing to be interviewed will be drawn in relation to variables such as age, frequency of drug use and sex work behaviour (using baseline questionnaire data) and levels of engagement with the intervention (using attendance data). Use of purposive sampling will aim to select interview participants that provide maximum variation in views and experiences. Interviews will be conducted face-to-face and written informed consent will be taken before starting the interview. Interview participants will be given a £20 high street shopping voucher as a thank you for their time. Participant interviews will be conducted with (1) service users that complete the 'Getting started' and 'Preparation for Recovery' groups and are not diagnosed with PTSD, (2) service users that complete the 'Stabilisation' group and treatment for complex PTSD and (3) service users that withdrew from any of the treatment groups. These interviews will consider and compare service user and service provider views, experiences, acceptability and costs of the intervention and suggested modifications to the intervention and study design. SSW interviews will explore initial impressions of the intervention, views on the recruitment strategy, factors influencing intervention attendance and experiences of the intervention including perceived benefits.

Service provider interviews will be conducted towards the end of the study. In addition to the topics above, these interviews will seek to understand operational issues of running the intervention, interagency working and general perspectives on delivering the intervention.

The sample sizes will be determined by the need to achieve data saturation, such that no new themes are emerging from the data by the end of data collection.[41] Interviews will be analysed in batches, and sampling will continue until no new themes are emerging from the interviews. The sample size of up to 20 service users and up to 15 service providers is expected to be sufficient to achieve this aim. With informed consent from participants, interviews will be recorded using a digital voice recorder, transcribed and anonymised to protect confidentiality.

## Data analysis
### Statistical analysis
All statistical analyses will be carried out according to the study analysis plan. The analysis plan details that we will conduct descriptive analyses using means, SD and non-parametric measures (where appropriate) to describe the characteristics of the participants and to analyse the feasibility and study process data. These will include but will not be limited to the number of participants approached to participate and their recruitment and retention, the number of participants partially and fully completing the intervention at each stage and

participant patterns of attendance. Resource use data collected on staff time and accommodation use will be multiplied by relevant unit cost data to generate a basic cost associated with provision of the intervention.

### Qualitative analysis
For the observations, the researcher will write detailed anonymised field notes, which will be transcribed for analysis. Interview audio files will be fully transcribed, anonymised and checked for accuracy. Observation field notes and interview transcripts will be imported into NVivo V.10 qualitative data analysis software to aid data management. Analysis will begin shortly after data collection starts and will be ongoing and iterative. Analysis will inform further data collection: for instance, analytic insights from data gathered in earlier interviews will help identify any changes that need to be made to the interview topic guide for use during later interviews. Thematic analysis,[42] using a data-driven inductive approach,[43] will be used to scrutinise the data in order to identify and analyse patterns and themes of particular salience for participants and across the data set using constant comparison techniques.[44 45] One researcher will lead the analysis, but other team members will independently code a subsample of transcripts, and all will meet to discuss the preliminary coding framework and themes, to ensure that the emerging analysis is trustworthy and credible and to maximise rigour.

## Patient and public involvement
Intervention and study design was developed based on input from the study PPI group that included women currently and previously involved in SSW and illicit drug use. The group convened before and during set up, contributed to the protocol development as well as the design of participant facing study documentation. Subsequent meetings have informed recruitment, topic guides, plain language study summary and plans for study dissemination. Ongoing PPI meetings will focus on troubleshooting issues identified during the study process and at the end of the study will focus on interpretation of results and dissemination methods.

## ETHICS AND DISSEMINATION
Ethical approval for the drug use in street sex workers DUSSK) study has been received.

## Summary of consent procedures
Eligible participants will provide fully informed, written consent to participate in the study at the time of screening and, if they are willing, consent to be contacted at a point later during the study about a subsequent qualitative interview. Written informed consent will be provided before starting the recorded qualitative interviews with participants and service providers. Qualitative researchers will observe treatment groups after obtaining the participants' verbal consent.

## Adverse events

Adverse events (AEs) and standardised operating procedures have been developed and will be followed by all researchers and service providers working on the study. Any unexpected AE defined as any untoward medical occurrence in a study participant to whom an intervention has been administered and serious AE (SAE) (defined below) will be reported by the researchers and service providers to the principal investigator who will keep records of each event to be monitored and reviewed at monthly Project Management Group meetings.

The principal investigator (NJ, a consultant in sexual health), study collaborator/designer (JM, a Professor, GP and expert in drug dependence and sex worker health inequality) and study coordinator (NMR) will assess the nature of reported AEs and SAEs for seriousness, causality and expectedness. Following the initial report, follow-up data may be requested by the study coordinator. All SAEs assessed to be related to the intervention and unexpected will be reported to the main Research Ethics Committee, Health Research Authority, the Sponsor and its research governance office, within 15 days of receiving notification of the SAE. If the individual affected is considered to be at ongoing risk, their caseworker will be informed.

## SAE definition

The definition of a SAE is any untoward and unexpected medical occurrence or effect in a study participant that is related to the intervention which: results in death; is life-threatening (refers to an event during which the participant was at risk of death at the time of the event, it does not refer to an event which might have caused death had it been more severe in nature); requires hospitalisation, or prolongation of existing hospitalisation; results in persistent or significant disability or incapacity or is otherwise considered medically significant by the investigator.

## Study sponsorship

The University of Bristol is the Sponsor for the study. Delegated responsibilities will be assigned to the University and NHS trusts taking part in this study. Collaboration for Leadership in Applied Health Research and Care (CLAHRC) West is responsible for, and administer, the financial aspects of the study. The study is open to inspection and audit by the University of Bristol under its remit as Sponsor.

## Dissemination

The study findings will be disseminated through publication in peer-reviewed open access journals as well as presentation at local and national conferences. We will make commissioners aware of our findings through meetings and circulation of appropriate materials highlighting the results. We will also ensure study participants, and members of the research population more widely, are aware of the findings through flyers and presentations. We will involve service users and our PPI group in all stages of dissemination and encourage them to copresent and contribute if they feel that is appropriate.

**Author affiliations**
[1]University Hospitals Bristol NHS Foundation Trust (UHBT), Bristol, UK
[2]Population Health Sciences, Bristol Medical School, University of Bristol, Bristol, UK
[3]The National Institute for Health Research Collaboration for Leadership in Applied Health Research and Care West (NIHR CLAHRC West), University Hospitals Bristol NHS Foundation Trust (UHBT), Bristol, UK
[4]The National Institute for Health Research Health Protection Research Unit in Evaluation of Interventions, University of Bristol, Bristol, UK
[5]Centre for Academic Primary Care, Population Health Sciences, Bristol Medical School, University of Bristol, Bristol, UK
[6]Health Economics at Bristol, Population Health Sciences, Bristol Medical School, University of Bristol, Bristol, UK
[7]Bristol Drugs Project, Bristol, UK
[8]Avon and Wiltshire Mental Health Partnership NHS Trust, Bristol, UK
[9]One25, The Grosvenor Centre, Bristol, UK

**Acknowledgements** The authors are extremely grateful to all the women who have participated in the study; all service providers and all other staff whose participation made this study possible. They would like to thank the PPI group for their time and thoughts. The authors are grateful to the following individuals who have helped the study with their time, expertise and support: Developing Health and Independence, Bristol Drug Project and One25 charities, Katie Warner, Lucy Pettler, Elaine Driver, Anna Smith, Rhea Warner, Jennifer Riley, Tracey Tudor, Sarah Shatwell, Katrina Turner, Hasina Downie, Jo Daniels, Jessica Munafo, Louisa Chowen and Faith Martin.

**Contributors** NJ, JM, NMR, JH, JMK, RP, SR and JC are responsible for the study design and collection of data. NMR, NJ and JH are responsible for study management and coordination. NJ, RP and JH drafted the paper. MT, DW and GN contributed to the design of the intervention. All authors read, commented on and approved the final manuscript.

**Funding** The research is supported by a National Institute for Health Research (NIHR) Clinic Trials Fellowship awarded to NJ, the National Institute for Health Research Collaboration for Leadership in Applied Health Research and Care West (NIHR CLAHRC West) at University Hospitals Bristol NHS Foundation Trust (UHBT) and Research Capability Funding awarded by UHBT. JMK is partly funded by NIHR Health Protection Research Unit in Evaluation of Interventions.

**Disclaimer** The views expressed are those of the authors and not necessarily those of the NHS, the NIHR or the Department of Health and Social Care.

**Competing interests** None declared.

**Patient consent** Obtained.

**Ethics approval** South West-Frenchay Research Ethics Committee (REC reference: 17/SW/0033; IRAS project ID: 220631).

**Provenance and peer review** Not commissioned; peer reviewed for ethical and funding approval prior to submission.

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
