## [Reviewer comments · BMJ Open]

ARTICLE DETAILS

TITLE (PROVISIONAL)	Drug use in street sex workers (DUSK) study protocol: A feasibility and acceptability study of a complex intervention to reduce illicit drug use in drug dependent female street sex workers
AUTHORS	Jeal, N; Patel, Rita; Redmond, Niamh; Kesten, Joanna; Ramsden, Sophie; Macleod, John; Coast, Joanna; Telfer, Maggie; Wilcox, David; Nowland, Gill; Horwood, Jeremy

VERSION 1 – REVIEW

REVIEWER	Amanda Roxburgh National Drug and Alcohol Research Centre, University of New South Wales
REVIEW RETURNED	09-Apr-2018

GENERAL COMMENTS	Review of Drug use in street sex workers (DUSK) study protocol It's really exciting to see that researchers are keen to continue development of interventions for street based sex workers who often don't have access to services and are often extremely stigmatised for the work they engage in. And fantastic to see that this sort of work is being funded and supported. I'm a little confused as the study dates appear to have passed which means that this protocol has already been applied, and the research project conducted. That being the case, I'll make my comments brief. Overall comments If this study has already been conducted then language should be changed to past tense. 1) There is a lot of literature suggesting that it is not good clinical practice to engage in trauma treatment with populations that are still at risk of being exposed to trauma – and these women most certainly will be in this situation for as long as they continue working in street-based markets. The authors need to address this in the protocol, and how they minimised the risks associated with this.2) It's not clear whether there were referral pathways for the women to see a clinical psychologist or obtain assistance
---

	should they need to at any point during the trauma treatment/research.  3) Can the authors please say a bit more about the evidence base for EMDR – perhaps deferring to gold standard treatment guidelines such as a Cochrane review. There has been a lot of debate about this technique in the literature and the mechanism that underpins its success. Addressing this is important. 4) Can the authors please also discuss the appropriateness of the use of EMDR for this group of women. Although it's seemingly less cognitively taxing than cognitive behavioural therapy, it does still rely on cognitive processing of the memories which is likely to be distressing for this group. 5) Can the authors also please refer to treatment guidelines with respect to how they chose the number of treatment sessions for each component of the study? Introduction The authors need to set up the rationale for conducting drug treatment and trauma treatment more clearly in the introduction. Why are they doing both? They just jump from high levels of drug use and high levels of mental health issues to development of the intervention. Drawing a bit more on the literature here would be good to set the scene for the reader. Drug Treatment Group Can the authors define 'evidence of life/drug use stability.' Screening for PTSD Can the authors be a bit more explicit about the PCL-5 – ie. As a screener for PTSD and also symptoms are assessed within the past month. Any discussion of PTSD should not be about 'experiencing PTSD', but rather currently experiencing PTSD symptoms, or screening positive for a provisional diagnosis of PTSD. Adverse Events This part of the protocol is written from the perspective of research governance and maintenance of relevant records, however, from a clinical perspective it's really important to have a protocol in relation to responding to psychological distress among the participants. These women are often highly marginalised and have many distressing things happening in their daily lives, that will occur above and beyond the research, and which will require an appropriate clinical response. Can the authors outline this a bit more explicitly and how they plan to or did deal with these situations as they arose.
--	--

REVIEWER	Sarah MacCarthy RAND, USA
REVIEW RETURNED	19-Apr-2018

GENERAL COMMENTS	BMJ Open This is a very important study. The manuscript is well written but lacks critical details that once addressed should be suitable for publication. Abstract  • Greatest risks to health compared to? • I don't understand the framing of the female SSW-only in the abstract – maybe just saying only drug dependent SSW in the treatment group (or something like that) but as currently written I find it confusing • Also, SSW only setting is unclear – meaning a clinic setting? A service center? On the street? It's hard to understand what you are talking about as currently written • Word count probably tight but if you could give parameters of treatment (x sessions for how long) • Abstract says feasibility and acceptability but title says feasibility only – please make consistent Strengths and limitations  • Considering clarifying that the systematic review of evidence was quantitative / empirical (mainly to be explicit about your consistent quant / qual focus) • I agree with your statement about efficacy but think that often readers (even educated ones!) get confused about feasibility vs efficacy so maybe just include a few words like "... about effectiveness or efficacy of the intervention are not possible at this stage due to the fact that x and y, however this study..." That way it reinforces that these are limitations of feasibility studies more generally not specific to your study. INTRODUCTION  • Say 'and' not 'but' when saying SSW experience worse health than • Consider if necessary to specify that you're talking about illicit drug use right from the beginning • 'cost effectiveness for female service users in general' what do you mean by this? I would not think that female service users would be more expensive than male service users? • 'related to their sex work history' such as ...? Just give one or two examples. In general – giving more brief / concrete examples would be helpful.
--

- By using the term robust and convincingly which make me think that there is something out there but that they weren't conclusive. I think helpful to name what is out there and why they weren't conclusive. That also will help establish the launching pad for why what you're doing is different / better.

DEVELOPMENT OF THE INTERVENTION

- Change first sentence to 'a novel intervention addressing the unique...'
- Pre-mainstream? What do you mean by this?
- Focus on PTSD means that you should include more PTSD specific stats up front in the intro rather than generically speaking to poor health outcomes. This will help better justify focus on PTSD rather than a range of negative health outcomes, or even other mental health outcomes.

METHODS AND ANALYSIS

- The goal is only to reduce illicit drug use? What about reduce levels of PTSD? Be more clear about your research question.
- Focused only on heroin or crack cocaine? So many other illicit drugs that could be included. Again, more stats needed up front to focus on these two drugs rather than including a broader range of drugs for inclusion criteria
- Intervention pathway – please be more specific about how you will adapt group allocation / re-allocation
- SSW only environment still unclear to me – on the street? In a local NGO? Something else? I need a clear picture of where the intervention is taking place
- Include table that lays out content of each of the sessions for the different groups – currently the session content is very generic
- Where will the PTSD treatment occur? How with such variable schedules will this be carried out? Incentives for participation? Payment for transport to sessions?
- High drop-out rates anticipated – what can you do up front to improve retention?
- Need cite to justify focus on 10. Even for feasibility study the lit I am familiar with suggests n=30 needed.
- A lot more baseline data needed for example history of violence (verbal, physical, mental), sexual abuse (childhood and adult) as well as more on demographics (education etc).
- Fidelity checks for delivery of intervention?
- Levels of sex work and levels of engagement with the intervention? Define what you mean by this.
- Include an interview scripts – right now topics are generic and more detail regarding specific questions for SSW vs providers needed
- You reference study analysis plan but instead should just include that here
- What about achieving kappa for qual analysis?

VERSION 1 – AUTHOR RESPONSE

Reviewer's comments	Author's response
Reviewer 1	
The abstract has dates that have already passed suggesting that the research has already been conducted. If this is the case then topline - even just numbers recruited should be reported. I'm a little confused as the study dates appear to have passed which means that this protocol has already been applied, and the research project conducted.	The paper was submitted before recruitment closed in keeping with requirements for the submission of a protocol paper. This study is currently ongoing with the intervention being delivered until December 2018. Our preference is to report recruitment numbers and the flow of participants throughout the study in a separate results paper once the feasibility study has completed.
If this study has already been conducted then language should be changed to past tense.	As stated above, this study has not been completed, so we have not changed the tense to the past tense at this time.
1) There is a lot of literature suggesting that it is not good clinical practice to engage in trauma treatment with populations that are still at risk of being exposed to trauma – and these women most certainly will be in this situation for as long as they continue working in streetbased markets. The authors need to address this in the protocol, and how they minimised the risks associated with this.	Literature that advises against engaging individuals in trauma-based therapy whilst still at risk of further trauma is based on what is most likely to be successful. Persons exposed to further traumatic events may be retraumatised. The SSW population will remain at risk of being exposed to trauma whilst still in the cycle of sex work to fund substance use to manage PTSD symptoms. Ideally, we would not be treating people at risk of further trauma but to uphold that would be to deny this treatment to all the SSW population. The protocol addresses how risks were minimised– clear support network for women with multiple points of access including ongoing SSW-only drug treatment groups which they have been attending over recent months so good therapeutic relationship with facilitators; each woman has a case worker from specialist sex worker support service who will receive feedback from the clinician delivering the EMDR if there are concerns; acute mental health services for women needing more intense support. This is reported on pages 8-11.
2) It's not clear whether there were referral pathways for the women to see a clinical	The women are able to see the clinical psychologist or request further assistance from

psychologist or obtain assistance should they need to at any point during the trauma treatment/research.	mental health services in line with routine practice if required. A sentence explaining this has been added on page 11.
3) Can the authors please say a bit more about the evidence base for EMDR – perhaps deferring to gold standard treatment guidelines such as a Cochrane review. There has been a lot of debate about this technique in the literature and the mechanism that underpins its success. Addressing this is important.	EMDR is recommended, alongside Trauma Focussed CBT, as a first line treatment for PTSD in the NICE Guidelines CG26. There are continued discussions on the mechanism that underpins the treatment and recent discussion has been around the hypothesis that EMDR has similarities with Slow Wave Sleep and its role in memory consolidation (Pagani et al. 2017). We have added this information on page 6.
4) Can the authors please also discuss the appropriateness of the use of EMDR for this group of women. Although it's seemingly less cognitively taxing than cognitive behavioural therapy, it does still rely on cognitive processing of the memories which is likely to be distressing for this group.	We acknowledge that this approach is unique and experimental, hence why this is a feasibility study to assess if this is an appropriate approach to take. As discussed in the NICE Guidelines, there is not good evidence for the use of non-trauma focussed interventions for the treatment of PTSD. If we are therefore to treat the PTSD we would use trauma focussed CBT or EMDR. EMDR was chosen because it does not require homework which may be a challenge for drug dependent SSWs who often have complex and unstable social backgrounds. It can be a relatively short course of treatment and is delivered on a one-to-one basis which is in keeping with the findings of a recent Cochrane review (Roberts et al. Psychological therapies for post-traumatic stress disorder and comorbid substance use disorder. Cochrane Database of Systematic Reviews 2016) which has now been added to the introduction on page 5. Processing of traumatic memories is likely to be distressing for any affected individual requiring this treatment, but the aim of the treatment is to reduce the associated distress to permit processing and appropriate storage of the memories. Any women undertaking this will be supported by their caseworker, the clinical psychologist undertaking the EMDR and the wider mental health support services team where needed.
5) Can the authors also please refer to treatment guidelines with respect to how they chose the number of treatment sessions for each component of the study?	The NICE Guidelines state that trauma focussed psychological therapy for PTSD should be 8-12 sessions of 90 minutes duration for a single event trauma, this has now been added to the introduction on page 6. There is a likelihood that

	the SSW participants have more complex problems than a single event trauma, for which the NICE Guidelines point out that the client may need more sessions. This is not necessarily the case, but should the participant require further treatment they will be assisted in accessing further treatment via local statutory services. If the 12 sessions have not been enough the participant is likely to have noticed enough improvement to encourage further help seeking.
Introduction The authors need to set up the rationale for conducting drug treatment and trauma treatment more clearly in the introduction. Why are they doing both? They just jump from high levels of drug use and high levels of mental health issues to development of the intervention. Drawing a bit more on the literature here would be good to set the scene for the reader.	Changes have been made to the Introduction to better highlight the reasons for doing both drug and trauma treatment, drawing more clearly on the literature on page 5.
Drug treatment group Can the authors define ‘evidence of life/drug use stability.’	Working closely with local drug service providers and the group facilitators criteria such as reliable group attendance (three out of four weeks) routine behaviours and interaction with drug treatment group facilitators, engagement and functioning in the group. As well as regular receipt of opiate substitution treatment. The authors are aware of resources such as the Cristo inventory to provide a measure of dependency-related problems but are of the opinion that for the purpose of assessing suitability for progression through the intervention that slightly different criteria may be required. As part of study data collection qualitative interviews with our intervention delivery partners will seek to clarify which assessment criteria group facilitators found most reliable in predicting readiness for transition to the next stage of the intervention. On page 9 we have expanded the description of this by explaining the study approach.
Screening for PTSD Can the authors be a bit more explicit about the PCL-5 – ie. As a screener for PTSD and also symptoms are assessed within the past month. Any discussion of PTSD should not be about ‘experiencing PTSD’, but rather currently	We have expanded the definitions and phrases as suggested, on page 10.

experiencing PTSD symptoms, or screening positive for a provisional diagnosis of PTSD.	
Adverse events This part of the protocol is written from the perspective of research governance and maintenance of relevant records, however, from a clinical perspective it's really important to have a protocol in relation to responding to psychological distress among the participants. These women are often highly marginalised and have many distressing things happening in their daily lives, that will occur above and beyond the research, and which will require an appropriate clinical response. Can the authors outline this a bit more explicitly and how they plan to or did deal with these situations as they arose.	When devising the protocol and study intervention we were very conscious of the high risk of adverse events (AE) and serious adverse events (SAE) in this population and spent time reviewing this thoroughly to ensure our Standard Operating Procedure addressed this. This was also reviewed by the Ethics committee and the Sponsor prior to submitting the protocol for Health Research Authority review. There are three clinicians on the study team (a Consultant in women's health specialising in hard to reach groups and the study's PI; a Professor, GP and expert in drug dependence and sex worker health inequality, and a consultant in in EMDR and clinical lead for the Trauma Service), to whom adverse and serious adverse events will/can be reported to. The AE and SAE reporting and action process is via the local hospital processes and the mental health services both in the hospital and the community are also on hand to facilitate any events. We have explained this in more detail on page 14, bearing in mind the journal's word limit.
Reviewer 2	
Abstract	
Greatest risks to health compared to?	We have added in 'compared to the general population' to this sentence.
I don't understand the framing of the female SSW-only in the abstract – maybe just saying only drug dependent SSW in the treatment group (or something like that) but as currently written I find it confusing	The intervention has been designed based on previous research to address the specific needs of female street sex workers. We have made it more explicit in the abstract that the intervention components are tailored for female street sex workers for example in relation to the drug treatment groups being held in the premises of a female street sex worker only charity.
Also, SSW only setting is unclear – meaning a clinic setting? A service center? On the street? It's hard to understand what you are talking about as currently written	We have added in "(a female sex worker charity premises)" into this sentence and this is explained further in the methods under 'Study Setting' on pages 7-9.
Word count probably tight but if you could give parameters of treatment (x sessions for how long)	The reviewer is correct in that unfortunately, due to other modifications, there is no further room to modify this in the abstract. However, these details are in the methods on pages 8-11.

Abstract says feasibility and acceptability but title says feasibility only – please make consistent	The title has been changed for consistency, as requested. We have modified this throughout the manuscript.
Strengths and limitations	
Considering clarifying that the systematic review of evidence was quantitative /empirical (mainly to be explicit about your consistent quant / qual focus)	Thank you for suggesting this. We have added in 'quantitative' to the second bullet point to address this on page 4.
I agree with your statement about efficacy but think that often readers (even educated ones!) get confused about feasibility vs efficacy so maybe just include a few words like "... about effectiveness or efficacy of the intervention are not possible at this stage due to the fact that x and y, however this study...." That way it reinforces that these are limitations of feasibility studies more generally not specific to your study.	Thank you for this suggestion. As suggested we have modified the bullet point by adding "...are not possible due to the study being a single arm feasibility study , however this study...." On page 4.
Introduction	
Say 'and' not 'but' when saying SSW experience worse health than	We have changed this as suggested on page 5.
Consider if necessary to specific that you're talking about illicit drug use right from the beginning	We have changed this as suggested in the first paragraph of the introduction to make it clear we are referring to illicit drug dependency.
'cost effectiveness for female service users in general' what do you mean by this? I would not think that female service users would be more expensive than male service users?	The study we reference found that in a mixed-gender drug treatment program women showed less improvement than men in the outcome of any substance use and to be more costly to treat. We have amended the sentence to improve clarity on page 5.
'related to their sex work history' such as ...? Just give one or two examples. In general – giving more brief / concrete examples would be helpful.	We have added to this sentence with "For example, feelings of stigmatisation from other male and female service users following disclosure of sex work and adverse interactions with previously known male service users potentially prevents SSWs from discussing unresolved trauma, undermining their engagement in treatment" on page 5.
By using the term robust and convincingly which make me think that there is something out there but that they weren't conclusive. I think helpful to name what is out there and why they weren't conclusive. That also will help establish the	Changes have been made to the introduction more clearly describing approaches used in previous SSW-targeted interventions aiming to reduce drug use. Also, clarification of why there was the need for this intervention and feasibility study on page 5/6.

launching pad for why what you're doing is different / better.	
Development of the intervention	
Change first sentence to 'a novel intervention addressing the unique...'	This has been changed on page 6.
Pre-mainstream? What do you mean by this?	We can see that this is not clear, this refers to this intervention being designed to happen prior to the usual drug treatment services usually conducted for any drug-dependent individuals in any given area. We have removed the word "pre-mainstream" and modified the sentence as follows "It was designed to occur prior to typical 'mainstream' drug treatment interventions (for both male and female drug dependent individuals) on page 6.
Focus on PTSD means that you should include more PTSD specific stats up front in the intro rather than generically speaking to poor health outcomes. This will help better justify focus on PTSD rather than a range of negative health outcomes, or even other mental health outcomes.	Changes have been made to the introduction to better explain the justification for the intervention PTSD focus.
Methods and analysis	
The goal is only to reduce illicit drug use? What about reduce levels of PTSD? Be more clear about your research question.	We have clarified throughout the paper that the aim of the intervention is to reduce levels of PTSD in order to support a reduction in illicit drug use in female drug dependent street sex workers.
Focused only on heroin or crack cocaine? So many other illicit drugs that could be included. Again, more stats needed up front to focus on these two drugs rather than including a broader range of drugs for inclusion criteria	The authors selected these drugs as the evidence base for these drugs is the most established and they are the most commonly used drugs among the population of interest. Sentences in the introduction have been modified to reflect this on page 5.
Intervention pathway – please be more specific about how you will adapt group allocation / re-allocation	This paragraph is attempting to highlight that, based on the individuals at the time of their movement through the study, the wider research team (with the support of the charity caseworkers, drug treatment professionals and mental health service team) will discuss and monitor how participants are and what they may benefit from. For example, as female sex-worker-only drug groups have not been held before, we may find that women benefit quicker from drug group sessions and may move quicker to the stabilisation group than initially

	anticipated. We have summarised this on page 8 by adding in “All partners will participate in monitoring how individuals and the wider group(s) are responding to the various aspects of the intervention, for example, we may find that women respond well to female, sex-worker only drug groups and develop stability behaviour more quickly than expected, in which case we may move them through the intervention quicker.”
SSW only environment still unclear to me – on the street? In a local NGO? Something else? I need a clear picture of where the intervention is taking place	We have detailed this further under ‘Study Setting’ on page 7. And have clarified throughout the paper that the location was a female sex worker charity’s premises
Include table that lays out content of each of the sessions for the different groups – currently the session content is very generic	On page 9 the drug treatment group content is outlined. The content is based on usual care. Prior to starting the feasibility study prescriptive /manualised group content was considered inappropriate due to the need to be responsive and develop group activities appropriate to the client group and numbers attending. The other unknown was how the unique group membership and setting would impact on ability of group attendees to discuss sex work experiences. This has previously described as impossible in mainstream settings and something that these groups were intended to accommodate. The group facilitators are familiar with both delivery of mainstream drug treatment groups and with this client group so are well-placed to undertake this task. This is aspect of the feasibility study will be captured during qualitative interviews with the group facilitators in order to guide development of a more prescriptive intervention approach for a subsequent pilot study. Due to word limits, we are not including this information in table format in the paper.
Where will the PTSD treatment occur? How with such variable schedules will this be carried out? Incentives for participation? Payment for transport to sessions?	PTSD treatment will take place in a clinical setting as stated on page 10, either in existing mental health service settings (offices) or in a rented room at a local GP practice. We have modified the “Treatment for PTSD” section by adding in “Treatment will occur in a clinical setting, for example, existing mental health service locations or local GP practice private room.”

High drop-out rates anticipated – what can you do out front to improve retention?	We are mindful of providing too many incentives for individuals and this could be costly and not practical in the future effectiveness trial, so this feasibility study will be assessing recruitment and aspects of retention to inform the full trial application.
Need cite to justify focus on 10. Even for feasibility study the lit I am familiar with suggests n=30 needed.	Our aim is to recruit 30 participants to the study, this will allow us to estimate the proportion of eligible people who are willing to participate, attrition rate and consider the practicalities of recruitment and delivering the intervention. This is reported on page 11. We have anticipated 30 - 60% attrition rate but this is speculative, so have removed this from the paper.
A lot more baseline data needed for example history of violence (verbal, physical, mental), sexual abuse (childhood and adult) as well as more on demographics (education etc).	As we are not assessing the effectiveness of the female sex worker only drug groups and trauma treatment in this study, we have kept the data burden on individual to a minimum, but this will be considered for the effectiveness trial in the future.
Fidelity checks for delivery of intervention?	The description of measures in the 'process evaluation' section will enable us to review intervention fidelity and we have added a sentence to reflect this at the end of this section on page 11 "These measures will also enable the assessment of the fidelity of the intervention as a whole against the proposed intervention pathway." As stated on page 8 under 'the intervention pathway' we have proposed a potential linear pathway, but this is unlikely to be adhered to strictly in practice, due to the nature of the participant's often chaotic lives. However, we will assess how closely the intervention adheres to this.
Levels of sex work and levels of engagement with the intervention? Define what you mean by this.	A purposive sample of those agreeing to be interviewed will be drawn in relation to variables such as age, frequency of drug use and sex work behaviour (using baseline questionnaire data) and levels of engagement with the intervention (using attendance data). We have clarified this on page 12.
Include an interview scripts – right now topics are generic and more detail regarding specific questions for SSW vs providers needed	The service user and provider interviews will explore the following topics: initial impressions of the intervention; views on the recruitment strategies; factors influencing attendance and how the intervention was received. In addition,

	service providers were asked to discuss operational issues, inter-agency working and to reflect on delivering the intervention. This detail is now included in the paper on page 12/13. Due to word limits, we are not including interview scripts in the paper.
You reference study analysis plan but instead should just include that here	The analysis plan is described but we concede this is not clear – we have modified this accordingly on page 13.
What about achieving kappa for qual analysis?	To enhance analysis and enable team discussion and interpretation, team members will independently code transcripts to ensure a wide range of codes are developed and any discrepancies discussed to achieve a coding consensus and maximise rigour. Using Kappa statistics to assess coding agreement is not necessary in qualitative research as formal interrater reliability is not a requirement. We will ensure the analysis is trustworthy, credible and rigorous by following a systematic coding process, documenting key decisions and using the collective insights of the team to interpret the data and reach consensus. To promote the trustworthiness of the coding procedures we will clearly articulate the coding procedures in the published manuscript of the findings.